# Modeling the effectiveness of olfactory testing to limit SARS-CoV-2 transmission

Daniel B. Larremore [1,2✉], Derek Toomre[3✉] & Roy Parker [2,4,5✉]

A central problem in the COVID-19 pandemic is that there is not enough testing to prevent infectious spread of SARS-CoV-2, causing surges and lockdowns with human and economic toll. Molecular tests that detect viral RNAs or antigens will be unable to rise to this challenge unless testing capacity increases by at least an order of magnitude while decreasing turn-around times. Here, we evaluate an alternative strategy based on the monitoring of olfactory dysfunction, a symptom identified in 76–83% of SARS-CoV-2 infections—including those with no other symptoms—when a standardized olfaction test is used. We model how screening for olfactory dysfunction, with reflexive molecular tests, could be beneficial in reducing community spread of SARS-CoV-2 by varying testing frequency and the prevalence, duration, and onset time of olfactory dysfunction. We find that monitoring olfactory dysfunction could reduce spread via regular screening, and could reduce risk when used at point-of-entry for single-day events. In light of these estimated impacts, and because olfactory tests can be mass produced at low cost and self-administered, we suggest that screening for olfactory dysfunction could be a high impact and cost-effective method for broad COVID-19 screening and surveillance.

[1] Department of Computer Science, University of Colorado Boulder, Boulder, CO, USA. [2] BioFrontiers Institute, University of Colorado Boulder, Boulder, CO, USA. [3] Department of Cell Biology, Yale University School of Medicine, New Haven, CT, USA. [4] Department of Biochemistry, University of Colorado Boulder, Boulder, CO, USA. [5] Howard Hughes Medical Institute, Chevy Chase, MD, USA. ✉email: daniel.larremore@colorado.edu; derek.toomre@yale.edu; roy.parker@colorado.edu

The COVID-19 pandemic has created a global public health crisis. Due to the fact that SARS-CoV-2 can spread from individuals with and without overt symptoms[1–3], there remains an urgent need to identify infected individuals prior to onward spread. To meet this need, large efforts are currently underway to develop, regulate, and mass-produce rapid, inexpensive, and effective screening tests for viral antigens that could be used repeatedly and at wide scale. However, an alternative approach is to utilize widespread screening of symptoms of SARS-CoV-2 infection, which could then stratify individuals with symptoms for follow-up molecular testing.

Fever was advanced early in the pandemic as a potential screening symptom but failed[4] because fever (≥38 °C) is (i) only present in 18–26% of COVID-19 cases[5–7], (ii) occurs in many diseases (e.g., Flu) and is not specific to COVID-19[8], and (iii) lasts only 1.5 days on average[5]. Nevertheless, temperature checks persist at many hospital entrances due to the speed and convenience of contactless thermometers and thermography, and because low cost yet effective alternatives are not widely available.

In contrast to fever, three key observations supported by a large and growing body of evidence indicate that olfactory dysfunction—hyposmia or anosmia—may be a superior screening and surveillance symptom. We briefly review this evidence, which may also be found in Supplementary Table 1. First, olfactory dysfunction is the best predictor symptom of COVID-19. Studies from both the U.S. CDC[9] and large cross-sectional questionnaires of thousands of subjects[10,11] have shown that OD predicts COVID-19 with an odds ratio of 10.4, a fourfold to tenfold higher association with COVID-19 than fever[9]. While OD is only weakly associated with influenza[12], it can be caused by other viral infections, head trauma, and chronic dementia[13].

Second, OD occurs in 76–83% of COVID-19 infections when measured with a quantitative olfactory test, such as a smell identification or threshold test[14–16]. Although reported prevalence of OD associated with PCR confirmed SARS-CoV-2 infection has varied widely in individual studies, pooled meta-analyses have established symptom prevalence estimates of 41% ($n =$ 24 studies[14]), 44% ($n = 34$ studies[15]), and 53% ($n = 19$ studies[16]) when OD was self reported. Strikingly, when OD was measured with a quantitative olfactory test, its estimated prevalence in meta-analyses increased to 83[14], 77[15], and 76%[16]. The stage or severity of disease may also play a role in the prevalence of OD. Several reports indicate that patients with mild COVID-19 had a high incidence of OD, ranging from 68% to 86%, but rates were far lower among severely ill patients[10,17]. However, this may be due to the fact that hospitalized patients are typically tested later in the course of infection, at which point OD symptoms may have largely resolved[18,19]. Furthermore, new reports using objective olfactory tests have found that 79% (22 of 28) of otherwise asymptomatic adults[20] and 86% of symptomatic children[11] showed partial loss of smell (hyposmia). To capture the entire range of prevalence estimates, the present study considers the implications of a range of COVID-19-associated OD prevalence values, ranging from 25 to 90%.

Third, onset of olfactory dysfunction may precede overt symptoms (e.g., difficulty breathing, cough, and diarrhea) by days[21–28], and COVID-19-induced olfactory dysfunction has been shown to last roughly 7 days in clinical studies[10,14,16,22], although in some cases may last for months[29]. Combined, olfactory dysfunction's prevalence, specificity, onset time, and duration have led us to hypothesize that, while it is not an overt symptom of COVID-19 and is underreported in self-reporting surveys[14,16], standardized olfactory dysfunction testing may be a valuable but underutilized screening and surveillance tool.

Recent modeling work has shown that, for COVID-19 screening via repeated molecular testing, test frequency and turnaround time are critical, while test sensitivity is secondary[30,31]. Standardized olfactory dysfunction testing may be sufficiently low cost to be performed frequently, and olfactory dysfunction testing can be self-administered in minutes without personal protective equipment (PPE). We therefore considered whether olfactory dysfunction could be effectively used in a similar repeated regimen to proposed molecular testing[30,32], and to what extent its effectiveness depends on its onset time, duration, and prevalence among those who are infected but do not experience overt COVID-19 symptoms. In this work, we define olfactory dysfunction to include any defect in olfaction that could be detected with a simple quantitative olfaction test, including the complete loss of smell (anosmia) and partial loss of smell (hyposmia).

## Results

**Overview of analyses.** We analyzed how screening for olfactory dysfunction could impact COVID-19 spread while varying the prevalence of olfactory dysfunction among infected individuals, its duration, the timing of onset, and the frequency of testing. In each case, we analyzed the impact of screening regimens in two manners.

In one set of analyses, we used a simple fully-mixed Susceptible-Exposed-Infectious-Recovered (SEIR) model representing a population of 20,000 people, similar to a large university setting, with a constant rate of external infection approximately equal to one new import per day[30]. Individual viral loads were simulated for each infection based on key features of latency, proliferation, peak, and clearance identified in the literature (Methods[30,33]). Infected individuals who scored positive for olfactory dysfunction were considered to be tested for SARS-CoV-2 by RT-PCR and were isolated if positive. To better model viral dynamics and behavior in the presence of overt and noticeable symptoms independent of olfactory dysfunction, 35% of modeled individuals had viral load trajectories with prolonged clearance times[33,34], and were modeled to self isolate within 0–2 days of peak viral load, independent of screening-related testing. Contact tracing was not included to more conservatively estimate the impacts of screening alone[35,36]. We used a value of $R_0$ of 1.5, to reflect accelerating but partially mitigated transmission.

In a second set of analyses, we simulated the viral loads, possible onset of olfactory dysfunction, and infectiousness curves, of 10,000 individuals and then examined how much infectiousness was removed from the population by different olfaction testing regimens. This allowed us to examine the results of modeling under a range of different screening strategies and olfactory dysfunction parameters by estimating the impact, in each case, on the reproductive number $R$.

In all analyses, we considered 80% of the population to participate in the screening protocol, examined performing olfactory testing either daily, every third day, or weekly, and infectiousness was taken to be proportional to the logarithm of viral load in excess of $10^6$ virions/ml[30].

One issue arising when screening for olfactory dysfunction is that defects in olfaction may be caused by other respiratory illness or early-stage neurodegenerative diseases[37,38]. COVID-19-independent olfactory dysfunction has been estimated to affect 3–25% of the general population, with the largest increase among older adults[37]. Higher specificity for COVID-19 olfactory dysfunction is possible, with some loss of sensitivity, by using test criteria that select for anosmia and severe hyposmia, excluding mild hyposmia and individuals with a stuffy or runny nose—a common symptom in other viral infections, but rare (2–4%) in COVID-19[39]. In consideration of this issue, we modeled that 20% of individuals would have a COVID-19-independent olfactory dysfunction, allowing us to examine the value of olfaction screening under

near-worst case real-world conditions[37,40,41]. Taken together, all simulations that follow assume that only 64% (80% of 80%) of individuals are both willing and able to participate in an olfactory screening program. Details of both models and parameters are fully described in Methods.

**Olfactory dysfunction screening can impact population spread.** We first examined how the prevalence of olfactory dysfunction during COVID-19 infection would impact its use in a repeated screening regimen. Olfactory dysfunction has been suggested to occur in 50–90% of COVID-19 infections for roughly 1 week[10,14,16,22]. Thus, we modeled olfactory dysfunction as a symptom able to be detected with repetitive olfactory testing in 25% (underestimate), 50% (low estimate), 75% (realistic), and 90% (high estimate) of infected individuals with an olfactory dysfunction duration of 7 days[10,22]. Although reports indicate that olfactory dysfunction is an early COVID-19 symptom[21–28], its precise timing relative to viral loads is unclear. Given this uncertainty, we initially modeled the average onset of olfactory dysfunction as occurring 2 days after viral loads reached detectable levels based on RT-PCR, consistent with studies indicating that onset of olfactory dysfunction precedes onset of overt symptoms[21–28].

We observed that screening for olfactory dysfunction daily (Supplementary Fig. 1A) or every third day (Fig. 1A) limited viral spread in simulations, provided symptom prevalence was >50%. Notably, when symptom prevalence was 75% or higher olfactory screening every third day was comparable in effectiveness to weekly RT-PCR testing with a 1-day turnaround time (Fig. 1; yellow dashed line) or weekly antigen testing (Fig. 1; red dashed line). At 50% symptom prevalence, viral spread was partially controlled by testing every 3 days (Fig. 1), and more effectively controlled with daily testing (Supplementary Fig. 1A). Weekly olfactory screening mitigated but failed to fully control outbreaks except when symptom prevalence was modeled at 90% (Supplementary Fig. 1B).

By estimating the reduction in the reproductive number $R$ for each scenario, we were able to perform direct comparisons of the predicted effectiveness of screening strategies across transmission scenarios (Fig. 1B). This illustrates how daily olfactory testing when symptom prevalence is 75% or greater is predicted to be slightly more effective than weekly antigen testing or RT-PCR with a 1 day turnaround time.

**Impact of timing of olfactory dysfunction onset.** Although olfactory dysfunction is an early symptom of COVID-19[21–25,27], there are limited data on the exact timing of symptom onset and its variability between individuals. Given this uncertainty, we performed additional analyses on how olfactory screening would be affected by the timing of symptom onset (Fig. 2). Assuming 80% participation in testing and 20% of the population having COVID-19-independent olfactory dysfunction, we considered the central estimates of 75% symptom prevalence and 7 days of duration, and varied the timing of symptom onset from 1 to 4 days after viral loads are detectable by RT-PCR.

We observed that daily, or every 3 days, olfactory testing was sufficient to keep viral infections from developing into an outbreak, provided that olfactory dysfunction typically occurs within 2 days of positivity by RT-PCR (Fig. 2). However, when symptom onset was 3 days after detectable viral loads, epidemic growth was poorly controlled with testing every 3 days, but could be controlled with daily testing (Fig. 2A and Supplementary Fig. 2A). By estimating the impacts of screening regimens on the reproductive number, we found comparable impact on transmission between weekly PCR and olfactory screening every 3 days (for onset 2 days after PCR positivity) or daily (for onset 3 days after PCR positivity). Even when symptom onset was 4 days after PCR positivity, viral spread was reduced by daily olfactory testing (Supplementary Fig. 2A). Weekly testing was largely ineffective when symptom onset was later than 1 day after viral loads reached detectable levels (Supplementary Fig. 2B).

**Impact of olfactory dysfunction duration.** Additional analyses argue that olfactory dysfunction screening could be an effective COVID-19 control mechanism even if the duration of olfactory dysfunction is shorter. For these analyses, we considered the case of 80% participation, 20% of the population having COVID-19-independent olfactory dysfunction, 75% symptom prevalence, symptom onset 2 days after viral loads become detectable by RT-PCR, and varied the symptom duration from 1, 3, 5, or 7 days.

We observed that testing every third day for olfaction controlled epidemic growth when the duration of olfactory dysfunction was 3–7 days (Fig. 3). Even if olfactory dysfunction lasted only 1 day—unlikely based on current observations—daily testing would nevertheless allow effective control. Weekly testing

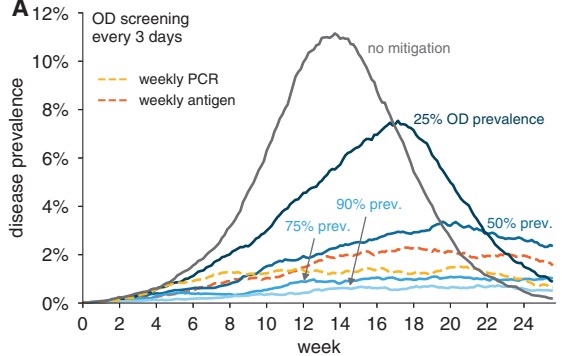
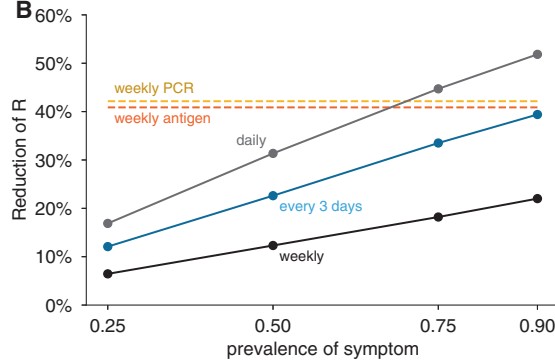

**Fig. 1 Impact of olfactory dysfunction prevalence on its effectiveness to limit viral spread. A** Examples of viral spread in a community of 20,000 individuals performing olfactory dysfunction (OD) screening every 3 days. No mitigation (black), Prevalence of symptom shown are: 25% (dark blue), 50% (medium blue), 75% (light blue), 90% (lightest blue). In this analysis, olfactory dysfunction is modeled to last 7 days and begin 2 days after viral levels reach ~1000 virions/ml. **B** Reduction of reproductive number $R$ with different testing regimens showing the impact of symptom prevalence with weekly (black line), every 3 days (blue line), or daily (gray line), testing for olfactory dysfunction. For comparison, weekly RT-PCR testing with a 1 day turnaround (dashed yellow line) and weekly antigen testing (dashed red line) are shown. We consider 80% participation in testing and 20% COVID-19-independent olfactory dysfunction in both panels.

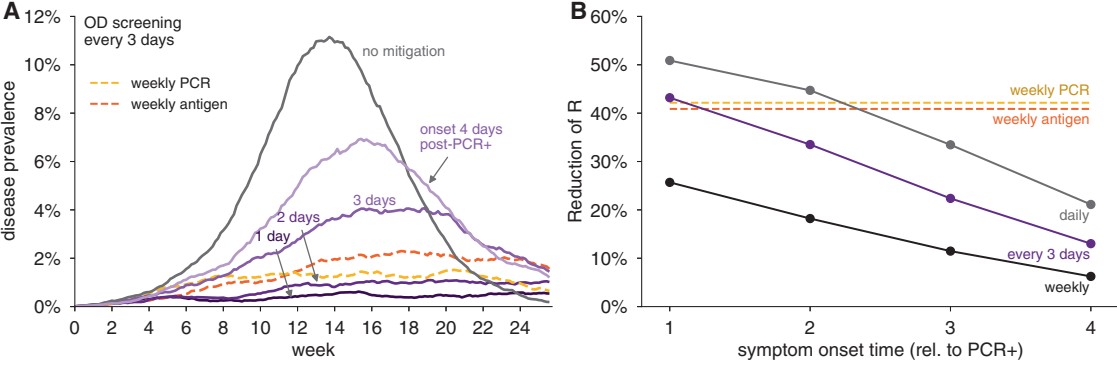

**Fig. 2 Impact of the timing of olfactory dysfunction onset on its effectiveness to limit viral spread. A** Examples of viral spread in fully-mixed community of 20,000 individuals performing olfactory dysfunction (OD) screening every 3 days. olfactory dysfunction is modeled to be present in 75% of infected individuals, and to last 7 days. Timing of olfactory dysfunction is varied from one to 4 days after viral loads reach ~1000 virions/ml (purple shaded lines as indicated). No mitigation is shown as black line. **B** Reduction of reproductive number *R* with different testing regimens showing the impact of timing of symptom onset with weekly (black line), every 3 days (purple line), or daily (gray line), testing for olfactory dysfunction. For comparison, weekly RT-PCR testing with a 1 day turnaround (dashed yellow line) and weekly antigen testing (dashed red line) are shown. We consider 80% participation in testing and 20% COVID-19-independent olfactory dysfunction in both panels.

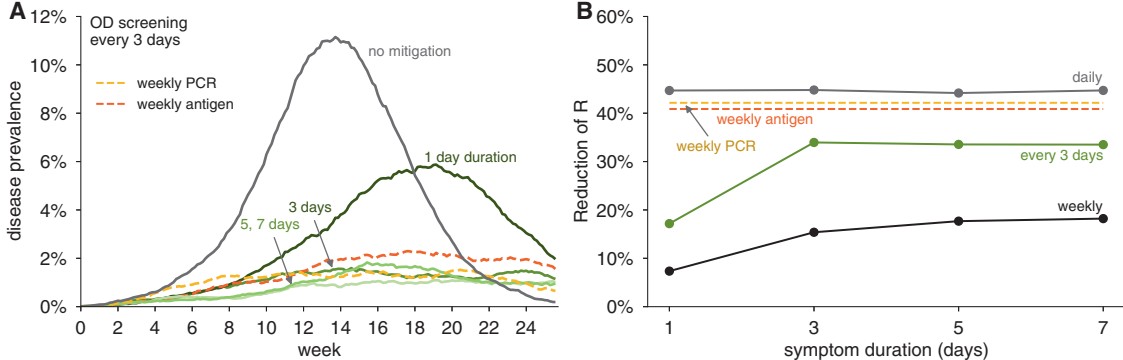

**Fig. 3 Impact of duration of olfactory dysfunction on its effectiveness to limit viral spread. A** Examples of viral spread in fully-mixed community of 20,000 individuals performing olfactory dysfunction (OD) screening every 3 days. Olfactory dysfunction is modeled to be present in 75% of infected individuals, and to begin 2 days after viral levels reach ~1000 virions/ml. Duration of olfactory dysfunction is varied from 7 days (lightest green), 5 days (light green), 3 days (green), to 1 day (dark green). No mitigation is shown as black line. **B** Reduction of reproductive number *R* with different testing regimens showing the impact of olfactory dysfunction duration with weekly (black line), every 3 days (green line), or daily (gray line), testing for olfactory dysfunction. For comparison, weekly RT-PCR testing with a 1 day turnaround (dashed yellow line) and weekly antigen testing (dashed red line) are shown. We consider 80% participation in testing and 20% COVID-19-independent olfactory dysfunction in both panels.

for olfaction was insufficient to maintain viral control (Fig. 3B and Supplementary Figs. 3B and 4L). To investigate interactions between onset timing and duration, we systematically modeled combinations of each, finding that testing for olfactory dysfunction every 3 days was effective, as assessed either by reductions in *R* (Fig. 4) or by viral containment in community simulations (Supplementary Fig. 4), provided that olfactory dysfunction lasted at least 3 days, and symptom onset was within 2 days of viral loads detectable by RT-PCR. Importantly, daily olfactory testing was sufficient to control viral spread even when the olfactory dysfunction onset was 3 days after detectable viral levels and lasted only for a single day (Supplementary Fig. 4A).

These analyses demonstrate that screening for olfactory dysfunction can be an effective control mechanism, even if average symptom duration is short, and symptom onset occurs within 3 days of detectable viral loads. More frequent testing for olfaction is required for effective control when the duration of olfactory dysfunction is short, and/or symptom onset is later in viral infection, with many scenarios providing reductions in the reproductive number equivalent to, or better than, weekly RT-PCR or antigen testing (Fig. 4).

**Screening for olfactory dysfunction to mitigate an outbreak.** To investigate whether olfactory screening could be effective at controlling an ongoing outbreak, we simulated epidemics in the fully-mixed SEIR model, such that a screening regimen began only when prevalence reached 2% of the population. We used central estimates of 75% olfactory dysfunction prevalence, symptom onset 2 days after detectable viral levels, and a symptom duration of 7 days. We again considered 80% participation in testing, with a further 20% of the population having COVID-19-independent olfactory dysfunction. We observed that screening daily or every third day was sufficient to bring the outbreak under control (Fig. 5), and under these conditions, would be similar to weekly RT-PCR or antigen testing.

**Olfactory dysfunction screening regimens are cost-effective.** To estimate the costs of modeled interventions, in each simulation we tracked the number of required olfactory dysfunction tests and follow up RT-PCR assays over a 120-day simulation (Table 1). In the repeated testing cases, we assumed individuals with COVID-19-independent olfactory dysfunction would not be

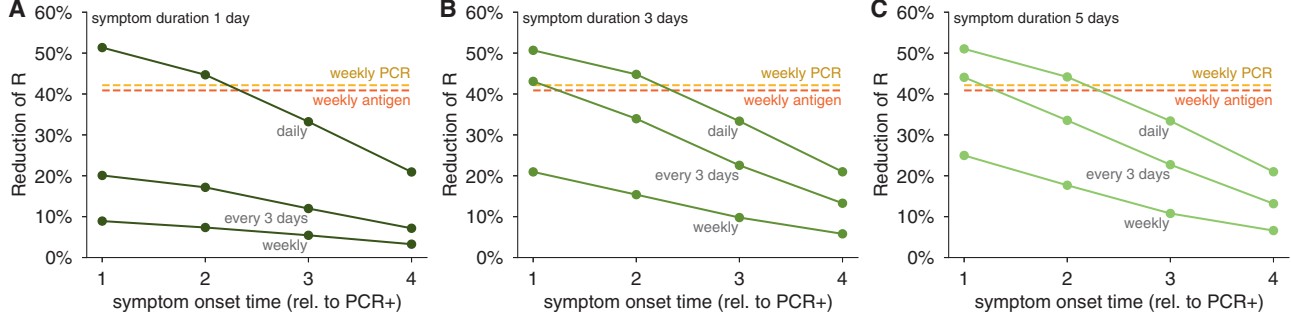

**Fig. 4 Impact of the timing of onset and duration of olfactory dysfunction on its effectiveness to limit viral spread.** Reductions of the reproductive number $R$ by testing daily, every third day, or weekly for olfaction dysfunction are shown for olfactory dysfunction lasting (**A**) 1 day, **B** 3 days, and (**C**) 5 days, for varying symptom onset times, relative to when viral levels reach ~1000 virions/ml. We consider 80% participation in testing and 75% prevalence of olfactory dysfunction as a COVID-19 symptom, and 20% COVID-19-independent olfactory dysfunction in all panels.

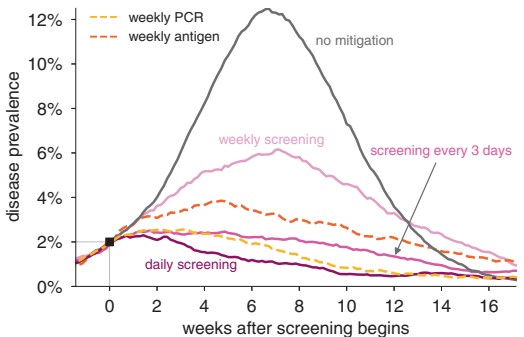

**Fig. 5 Impact of olfactory dysfunction screening on an ongoing viral outbreak.** Examples of the impact of screening programs based on olfactory dysfunction on controlling an ongoing outbreak. Viral spread in a fully-mixed community of 20,000 individuals was allowed to proceed until prevalence reached 2%, at which point olfactory dysfunction (OD) screening was initiated either daily (dark red), every 3 days (red), or weekly (pink). For comparison, weekly RT-PCR testing with a 1 day turnaround (dashed yellow line) and weekly antigen testing (dashed red line) are shown. Olfactory dysfunction is modeled to be present in 75% of infected individuals, last 7 days, and begin 2 days after viral levels reach ~1000 virions/ml. Our modeling assumed that 80% of people would opt to participate in testing and that 20% of people would be unable to participate due to COVID-19-independent olfactory dysfunction. Scenario without mitigation is shown for comparison (black).

tested by RT-PCR, and therefore the estimated costs of RT-PCR are minimum estimates. We found that a mitigation strategy based on weekly RT-PCR assays with 80% participation would reduce $R$ by 42% but cost $5.3 million in tests, even at the low rate of $20 per RT-PCR test ($100 is more typical). A similar weekly screening program using $10 antigen tests would reduce R by 41% and cost $2.7 million. However, mitigation based on olfactory screening every 3 days with follow-up RT-PCR would cost $130,000 in tests, assuming $0.25 per olfactory test, and reduce $R$ by (34%; see Table 1). By comparing the cost per percentage point reduction in $R$, olfactory screening was 16× or 31× more effective than antigen or PCR screening, respectively. Cost estimates do not include costs of staffing, PPE, or sample collection and transport which may be required for testing via RT-PCR.

**Olfactory dysfunction as a point of entry screening tool.** Due to the immediate results of currently available olfactory dysfunction tests, we also examined the conditions under which they would be useful as point-of-entry screening tools, analogous to screening of airline passengers or individuals entering a social event with rapid

antigen tests. We observed that screening for olfactory dysfunction removed same-day infectiousness in proportion to the fraction of individuals showing that symptom, and was more effective the earlier symptom onset occurred (Supplementary Fig. 5A). Symptom screening is ~80% effective when 90% of the infected individuals show the symptom (Supplementary Fig. 5A). Notably, even with only 50% of infected individuals showing olfactory dysfunction over 40% of infectiousness was removed provided symptom onset occurred no later than 2 days after sufficient viral load to be detected by RT-PCR testing (Supplementary Fig. 5A). In contrast, a symptom similar to fever, which is short lived and seen in a small number of infected individuals[5–7], removes at most 20% of infectiousness (Supplementary Fig. 5A).

One limitation of using olfactory dysfunction as a single step screen for COVID-19 is the prevalence of non-COVID-19 related olfactory dysfunction, often estimated at 3-5% of the general population[37,40,41], but with higher rates among particular groups, including older adults[37] and young children[42,43]. To address this issue, we considered a two-step screening procedure in which those failing an olfaction test would be given a reflex rapid antigen test, allowing those with COVID-19-independent olfactory dysfunction to enter the event. This two-step screening procedure was as effective as the one-step olfaction-only procedure, was cost effective (~$0.75 to $2.33 per person), and reduced false positives that were inappropriately denied entry compared to either test alone. The rapid turnaround time of both olfactory dysfunction testing and rapid antigen testing, suggests that this two-step approach would be effective and superior to an olfaction-only screening procedure (Supplementary Fig. 5B, C).

## Discussion

An important contribution of this modeling is to demonstrate that monitoring of olfactory dysfunction could be effective at controlling the spread of SARS-CoV-2. Our analysis, and data in the literature, argue that olfactory dysfunction during COVID-19 meets all the necessary criteria of prevalence and specificity, timing, and duration for being effective in pandemic control. First, our modeling showed that the estimated 75% prevalence of olfactory dysfunction among those with and without other overt COVID-19 symptoms[14,16,20] was sufficiently high to have a substantial impact in screening regimens (Fig. 1). Second, we showed that screening would be effective even if onset of olfactory dysfunction is 2 days after detectable levels of virus by RT-PCR, with increasing predicted effectiveness with earlier onset times (Fig. 2). To date, there have been few prospective studies of loss of smell in COVID-19 patients, but existing evidence indicates an earlier onset of olfactory dysfunction than overt symptom like fever and headache[21–24,27]. Determination of the average timing

**Table 1 Cost effectiveness of repeated screening programs.**

| Screening regimen | OD tests | PCR/Ag tests | Cost (millions) | Reduction in $R$ | Cost effectiveness |
|---|---|---|---|---|---|
| OD daily | 1,531,351 | 151 | $0.39 | 44.7% | 115.9 |
| OD every 3 days | 509,360 | 314 | $0.13 | 33.5% | 250.7 |
| OD weekly | 213,963 | 457 | $0.06 | 18.2% | 290.7 |
| PCR every 3 days | – | 633,565 | $12.67 | 68.4% | 5.4 |
| PCR weekly | – | 265,695 | $5.31 | 42.1% | 7.9 |
| Ag every 3 days | – | 632,914 | $6.33 | 67.6% | 10.7 |
| Ag weekly | – | 268,088 | $2.68 | 40.9% | 15.2 |

Numbers of olfactory dysfunction (OD) tests and PCR tests ordered are translated to total costs for a 120 day testing program at a rate of $0.25 per OD test, $10 per antigen (Ag) test, and $20 per PCR test in simulations of a population of 20,000 individuals with $R_0 = 1.5$, Cost effectiveness is calculated as the percentage reduction in $R$ per million dollars. Individuals positive for olfactory dysfunction received a confirmatory PCR test. Olfactory dysfunction is modeled to be present in 75% of infected individuals, last 7 days, and to begin 2 days after viral loads reach 1000 virions/ml, with 20% of individuals experiencing COVID-19-independent olfactory dysfunction. We consider 80% participation in testing of any type.

of olfactory dysfunction relative to viral load in COVID-19 patients will be critical to implementing effective screening protocols. Third, our analysis suggests the even if the duration of olfactory dysfunction is as short as 1 day (the exact duration and its variability in infected individuals with no other overt symptoms is unknown), frequent monitoring for olfaction can still be effective for epidemic control (Fig. 3). Our simulations suggest effective epidemic mitigation can be achieved under all conditions with daily testing; with longer durations and earlier symptom onset, epidemics can be controlled with testing every 3 days.

Olfactory screening is inexpensive and scalable, two critical factors for large population-scale testing which differentiate olfactory screening from alternatives like antigen testing and RT-PCR. Because olfaction tests are inexpensive (we estimate $0.25/test) and uncomplicated, frequent repeated tests can feasibly be carried out by a large fraction of the population. For comparison, to test 50 million people/day (or the entire US population weekly) PCR tests costing $50 each would cost $2.5 billion per day, $10 antigen tests would still cost $500 million per day, while an olfactory test would cost $12.5 million per day—between 40× and 200× less. Moreover, because olfaction can be self-tested, there is no need for the logistics of sample collection and transport, which can further reduce costs. Nevertheless, both RT-PCR and antigen testing have been successfully used in screening and surveillance applications, and olfactory screening should be considered as a complement to, not a replacement for, existing successful programs.

A potential advantage of olfactory screening is its ability to be scaled. Paper-based olfactory dysfunction tests can be printed on industrial printers and a single commercial printing facility can produce over 50 million tests daily (D.T., communication). Simple olfaction tests linked to a mobile phone app have already been developed, are FDA registered, and are under consideration for Emergency Use Authorization (EUA) by the U.S. FDA for COVID-19 applications, as of March 2021. In contrast, March 2021 PCR testing maximum capacity was approximately 2.1 million tests/day[44]. Moreover, instant olfactory symptom reporting via a testing app could provide centralized surveillance and early warning of new outbreaks[24]. A facile test may be especially valuable in low and middle-income countries where access to complex molecular testing is cost prohibitive.

Our study is subject to a number of limitations. First, our modeling assumed that olfaction tests would not only correctly identify those with olfactory dysfunction, but would remain equally sensitive during regular use. In practice, however, sensitivity could increase or decrease over time as test takers better learned to differentiate the suite of test scents. Second, we assumed that no individual would intentionally fail a test–an issue only averted with reflex molecular testing. Finally, we assumed that olfactory dysfunction revealed by a test would lead to

isolation, but this assumption is unlikely to be valid for essential workers, or those uninterested in adherence to an isolation protocol. These limitations highlight the need to develop, deploy in longitudinal trials, and refine tests for olfaction that can be mass produced at low cost and self-administered.

There are three additional points to consider in implementing olfactory screening for COVID-19 control. First, when a new community is subject to olfactory screening, there will be an initial surge of individuals identified with olfactory dysfunction both due to undetected COVID-19 infections, and COVID-19-independent olfactory dysfunction (anosmia and hyposmia). This will require the ability to handle a reflexive surge in molecular testing during the initial phase. However, even in the absence of follow-up molecular testing, unnecessary quarantines of those without COVID-19 (false positives) should be weighed against blanket non-pharmaceutical interventions such as lockdowns and curfews. A second issue is that increased anosmia and hyposmia in older adults[37,40,41] will mean that olfactory screening will be most effective with younger adults, including college students. Finally, a critical public health issue inherent to any COVID-19 monitoring/surveillance mechanism is that no testing strategy identifies and isolates all the COVID-19 infected individuals, both from false negatives, and from new import of infections into a community. Therefore, any COVID-19 screening regimen must complement and not replace existing viral mitigation mechanisms such as mask wearing and social distancing.

## Methods

**Individual infection model.** Each individual infection consists of four key elements: (1) a viral load trajectory which charts the measurable concentration of virus over the course of infection, (2) the presence of symptoms so noticeable that they cause a change in behavior, i.e., symptom-driven self-isolation, (3) the presence of symptoms identifiable by a screening test, such as olfactory dysfunction or low fever, and (4) an infectiousness trajectory, which is related to both viral load and behavior. Each of these components is described in detail below.

When a positive screening test—virological or symptom-based—provides a positive test result during the individual's infectious window, prior to any self-isolation, that person is assumed to isolate for the remainder of the infection. Thus, depending on the individual's viral load, infectiousness, timing of possible test results, and timing of possible self-isolation, a screening test may or may not cause an individual's isolation and concomitant decrease in circulating infectiousness.

*Viral load trajectories.* Viral loads were drawn from a simple, previously published viral kinetics model which captures four key aspects of a SARS-CoV-2 infection[30]: (a) a variable latent period, (b) a rapid proliferation phase from the lower limit of PCR detectability to a peak viral load, (c) a slower clearance phase, and (d) prolonged clearance for symptomatic infections vs asymptomatic infections. These dynamics were based on a growing body of literature that includes longitudinal repeated PCR sampling of symptomatic and asymptomatic individuals, in both prospective and non-prospective contexts. The detailed studies on which the viral load was based were previously reviewed in [30], but have since been further supported through a prospective longitudinal study[33] which further refined the proliferation phase and differential clearance rates based on symptoms.

To summarize the model of ref. [30], with slight modifications to reflect knowledge gained from ref. [33], $\log_{10}$ viral loads were approximated by a

continuous piecewise linear "hinge" function, specified uniquely with three control points: $(t_0, 3)$, $(t_{peak}, V_{peak})$, $(t_f, 6)$. The first point represents the time at which an individual's viral load first crosses $10^3$, and becomes detectable via PCR, with $t_{PCR}$ ~ unif[2.5, 3.5], measured in days since exposure. The second point represents the peak viral load. Peak height was drawn $V_{peak}$ ~ unif[7, 11], and peak timing was drawn with respect to the start of the proliferation phase, $t_{peak} - t_{PCR}$ ~ 0.5 + gamma(2.5) with a maximum of 4. The third point represents the time at which an individual's viral load crosses beneath the $10^6$ threshold, at which point viral loads no longer cause active cultures in laboratory experiments[34,45–47]. For asymptomatic infections, this point was drawn with respect to peak timing, $t_f - t_{peak}$ ~ unif[4, 8]. For overtly symptomatic infections, a symptom onset time was first drawn with respect to peak timing, $t_{symptoms} - t_{peak}$ ~ unif[0, 2], and then the third control point was drawn with respect to symptom onset, $t_f - t_{symptoms}$ ~ unif [4, 8]. Thus, overtly symptomatic trajectories are systematically longer, in both duration of infectiousness and duration of viral shedding[33]. In simulations, each viral load's parameters were drawn independently of others, and the continuous function described here was evaluated at 21 integer time points, representing a 3-week span of viral load values.

*Overt symptoms causing self-isolation.* For individuals with overtly symptomatic infections (see description of Viral Load Trajectories), symptom onset at $t_{symptoms}$ caused self-isolation on the following day. Overt symptoms were assumed to be present in 35% of individuals, and were assumed to be present independently of olfactory dysfunction and low-grade fever, which are considered separately because they typically do not lead to behavior change.

*Analytical sensitivity of RT-PCR and antigen testing.* We assume an analytical sensitivity of RT-PCR of $10^3$ RNA copies/ml, consistent with typical performance. We assume an analytical sensitivity of antigen tests equivalent to $10^6$ RNA copies/ml, consistent with independent field evaluations of the Abbott BinaxNOW test[48]. When viral loads exceeded their respective limits of detection, we assume that these molecular tests correctly identify infected individuals.

*Symptoms identifiable by screening.* Symptoms such as olfactory dysfunction or low-grade fever were modeled using three variables, which controlled the prevalence $\theta$, onset time $t_{onset}$, and duration $d$ of symptoms. Onset times were chosen relative to detectability by PCR, such that $t_{onset} = t_{PCR} + k$, with $k = -2, -1, ..., 3$ evaluated in the text. Note that the prevalence parameter $\theta$ incorporates both the presence and the detectability of symptoms. As noted above, the prevalence of identifiable symptoms via this mechanism was considered independently of the overt symptoms that lead to self-isolation.

*Infectiousness.* Infectiousness $F$ was assumed to be directly related to viral load $V$ in that each individual's relative infectiousness was proportional to the $\log_{10}$ of viral load's excess beyond $10^6$, i.e., $F \propto \log_{10}(V) - 6$. For individuals in isolation, either following a positive screening test or following the appearance of overt symptoms (at $t_{symptoms}$), infectiousness was set to zero. In all simulations, the value of the proportionality constant implied by the infectiousness function was chosen to achieve the targeted value of $R_0$ for that simulation, and confirmed via simulation as described below.

**One-shot, two-shot, and repeated screening.** For an individual with viral load trajectory $V$ and infectiousness $F$ over the simulated 21 days of infection, proliferation, and clearance, testing was implemented on a specified day or on a schedule, as follows. First, each individual's viral load $V$ was drawn as described above, with 35% of individuals receiving an overtly symptomatic trajectory and 65% receiving an overtly asymptomatic trajectory. Each individual was also assigned screenable symptoms with probability $\theta$, lasting from day $t_{onset}$ through $t_{onset} + d$, and no screenable symptoms otherwise.

*One-shot and two-shot screening.* For disease transmission dynamics scenarios, on the day of one-shot screening, each individual was evaluated to determine whether they would receive a positive test result that day (symptom screening) or the next day (virological screening). Thus, the timing of each individual's viral load, symptom status, and infection status were determined by the dynamics of the simulation, described below. For non-dynamic scenarios, a statistical sample of 10,000 infected individuals were considered such that the timing of the screening test was equiprobable on each day of infection, with symptom screening and virological results returned on the same day and next day, respectively. Olfactory dysfunction tests were assumed to have a specificity of either 0.8 or 0.96, to explore the impact of vary rates of COVID-19-independent olfactory dysfunction.

In two-shot screening scenarios, individuals with positive symptom screening tests were referred to a reflex point-of-care rapid diagnostic test with an assumed limit of detection $L$. If $V(t) > L$ on that day, they received a positive result and isolated, but otherwise, they received a negative result and were allowed to enter the imagined event. Rapid diagnostic tests were assumed to have a specificity of 0.025. False positives by olfactory dysfunction test and by rapid diagnostic test were assumed to be statistically independent of each other, such that the specificity of the two-shot screening was 0.001.

*Repeated screening.* Based on a schedule of testing each person every $D$ days, if an individual happened to be tested by a symptom screening test on a day when their symptoms were present, their positive result would cause them isolate that day, without delay. Similarly, if an individual happened to be tested by a virological test on a day when their viral load exceeded the limit of detection of the test ($V(t) > L$), their positive result would cause them to isolate, but with a 1 day delay in virological test results. Each person was deterministically tested exactly every $D$ days, but testing days were drawn uniformly at random such that not all individuals were tested on the same day. Values of $D$ were 1, 3, or 7. In repeated screening scenarios, olfactory dysfunction tests were assumed to have a specificity of 1, reflecting a steady-state assumption that individuals would rapidly determine that their olfactory dysfunction was not COVID-19-related.

**Disease transmission model.** A fully-mixed model of $N = 20,000$ individuals with all-to-all contact structure, zero initial infections, and a constant $1/N$ per-person probability of becoming infected from an external source was used to simulate SARS-CoV-2 dynamics, based on a typical compartmental framework (e.g., as in[30]) but with modifications for symptom screening. These models tracked discrete individuals who were Susceptible ($S$), Infected ($I$), Recovered ($R$), Isolated ($Q$), and Self-Isolated ($SQ$) at each discrete 1-day timestep. Upon becoming infected ($S \rightarrow I$), a viral load trajectory $V(t)$ was drawn which included a latent period, growth, and decay as described above. For those chosen to have non-overt symptoms, the timing and duration were fixed, according to the choice of parameters and the $t_{PCR}$ of the viral load. Thus, each day, an individual's viral load trajectory was used to determine whether their diagnostic test would be positive if administered, as well as their infectiousness to susceptible individuals; The timing, duration, and prevalence of symptoms was used similarly to determine whether a screening test would be positive if administered.

*Participation in testing.* 20% of individuals were, in some simulations, selected to refuse testing. Testing refusal—or its complement, participation—was determined at random using a specified refusal rate, at the initialization of each simulation, and was unchanged for the duration of each simulation. In all cases except for the one-shot and two-shot screening scenarios, we considered an 80% participation rate (i.e., 20% refusal rate).

Some individuals who are interested in testing (i.e., they do not refuse a testing regimen) may nevertheless be unable to effectively participate due to existing full or partial loss of olfaction[37]. To conservatively model the impact of olfactory screening, particularly among older adults, we considered scenarios where either 5% of individuals or 20% of individuals, as noted throughout the text, had COVID-19-independent olfactory loss which rendered them unable to participate in testing programs.

Taken together, these two participation rates mean that for repetitive screening scenarios and simulations we assumed that only 80% × 80% = 64% of the population could effectively participate in a screening regimen. In one-shot and two-shot screening scenarios, only the inability to participate was considered, such that the specificity of the olfactory screening was either 95% or 80%.

*Isolation due to symptom screening or virological testing.* On the specified day(s) of testing, each infected person was evaluated to determine whether their viral load ($V(t) > L$ in the case of a virological test) or symptoms (presence/absence for that individual on that day, in the case of a symptom screening test) would cause a positive test result. Positive results caused isolation ($I \rightarrow Q$) with no delay or with a 1-day delay for symptom screening or virological tests, respectively.

*Self-isolation and recovery.* 35% of individuals self-isolate on the day of symptom onset, which occurs 0–3 days after peak viral load (see above), to mimic overt symptom-driven isolation ($I \rightarrow SQ$). Thus, presymptomatic individuals were isolated prior to symptom onset only if they received positive test results. When an individual's viral load dropped below $10^3$, that individual recovered ($I, Q, SQ \rightarrow R$).

*Transmission, population structure, and mixing patterns.* Simulations were initialized with all individuals susceptible, $S = N$. Each individual was initially chosen to either participate in testing or refused testing, as described above, independently with a probability specified per-simulation. Each individual was chosen to be overtly symptomatic independently with probability 0.35. Both participation/ refusal and overt symptoms were assumed to be persistent through the simulation, per person. If repeated testing was to be performed, each individual's first test day (e.g., the day of the week that their weekly test would occur) was chosen uniformly at random between 1 and $D$. Relative infectiousness was scaled up or down to achieve the specified $R_0$ in the absence of any testing policy, but inclusive of any assumed self-isolation of overt symptomatics.

In each timestep, those individuals who were marked for testing that day were tested. Individuals receiving a positive test result that day, after delay of virological test results, were isolated, $I \rightarrow Q$. Overtly symptomatic individuals whose viral load had declined relative to the previous day were self-isolated, $I \rightarrow SQ$. Next, each susceptible individual was spontaneously (externally) infected independently with probability $1/N$, $S \rightarrow I$. Then, all infected individuals contacted all susceptible

individuals, with the probability of transmission based on that day's viral load $V(t)$ for each person infectiousness function described above, $S \to I$.

To conclude each time step, individuals' viral loads and symptoms were advanced to the next time step, with those whose infectious period had completely passed moved to recovery, $I, Q, SQ \to R$.

*Ongoing screening vs outbreak mitigation scenarios.* In ongoing screening scenarios, simulations with $R_0 = 1.5$ and the constant rate of external infection were conducted with screening beginning, as described above, starting on the first timestep. In outbreak mitigation scenarios, simulations were identical except that no screening was performed until disease prevalence in that time step reached 2% of the population (400 individuals).

*Calibration to achieve targeted* $R_0$ *and estimation of* $R$. As a consistency check, each simulation's $R_0$ was confirmed to ensure that simulations were properly calibrated to their intended values. Note that to vary $R_0$, the proportionality constant in the function that maps viral load to infectiousness need only be adjusted up or down. In a typical SEIR model, this would correspond to changing the infectiousness parameter which governs the rate at which $I$-to-$S$ contacts cause new infections $\beta$.

For the fully-mixed model, the value of $R_0$ was numerically estimated by running single-generation simulations in which a 50 infected individual were placed in a population of $N - 50$ others. The number of secondary infections from those initially infected was recorded and used to directly estimate $R_0$.

Estimations of $R$ proceeded exactly as estimations of $R_0$ for both models, except with interventions applied to the viral loads, symptoms, and therefore the dynamics.

**Sample sizes.** Means (specifically, proportions) were estimated from 10,000 simulations, a number chosen so that standard errors would be <0.5% (0.005).

**Reporting summary.** Further information on research design is available in the Nature Research Reporting Summary linked to this article.

## Code availability

All code needed to evaluate the conclusions in the paper are present in the paper and/or the Supplementary Materials, and open-source code (Python 3.7.4) is available[49].

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

## Acknowledgements

The authors thank Yonatan Grad and A. Marm Kilpatrick for valuable feedback. This work was supported by the Howard Hughes Medical Institute (Roy Parker).

## Author contributions

D.B.L., D.T., and R.P. conceived and designed the study. D.B.L. performed the computational modeling. D.B.L., D.T., and R.P. wrote the manuscript.

## Competing interests

D.B.L. is a member of the scientific advisory board of Darwin BioSciences. D.T. declares competing interests as a founder of olfactory test company (u-Smell-it™LLC) and for EU patent (DM/212486) and pending US patents (29,743,100 and 29,750,313) that relate to an inexpensive multi-odorant olfactory test and software interface. R.P. declares no competing interests.
