## [Peer Review File · Nature Communications]

Reviewers' Comments:

Reviewer #1:

Remarks to the Author:

Dear authors and editors, thank you very much for the chance to review this report.

This is an interesting modelling study assessing the effect of frequent olfactory testing on infections and Rt. It is well and concisely written and captures some of the important parts of olfactory testing in COVID.

I have a general comment: I am not certain about the high prevalence of olfactory symptoms/positive olfactory dysfunction tests assumed in those with SARS-CoV-2 infections, but this is a crucial assumption underlying the conclusions of the paper.

As far as I understand the meta-analyses presented as references for the current estimates provide prevalence of anosmia mostly in those patients notified in cases and often from highly symptomatic patients such as hospital patients and not in those infected (detected via populationbased studies). In my opinion however the actual prevalence of those infected should be used as a modelling basis – which should be meta-analysed from population based studies (seroprevalence or other). I have not found such a meta-analysis but from our own data would assume that prevalence then would be lower. Also, in German contact tracing data (so notified cases) anosmia is present in 21%

(https://www.rki.de/DE/Content/InfAZ/N/Neuartiges_Coronavirus/Situationsberichte/Jan_2021/2021-01-19-de.pdf?__blob=publicationFile) , which of course may be an underestimate. If that was the case, sensitivity of olfactory tests is far below sensitivity of current antigen or PCR tests.

Currently the lower bound of sensitivity analyses provided by the authors is at 25% prevalence and does not show an effect (which would change conclusions of the paper if this was the more probably prevalence of anosmia in all infected) so I would ask authors to please

- perform/reference a meta-analysis/studies of anosmia prevalence in population based studies and parametrize the model/ change sensitivities accordingly
- change parameters and conclusions accordingly

As a second general comment I am not convinced by the used comparison to weekly PCR and would like to see additional comparison to higher frequency use of antigen tests/pooled PCR. This is important also as the authors do point out the low positive predictive value of olfactory testing because of prevalent non-COVID related olfactory symptoms in the population.

Introduction:

- please contrast the use of olfactory dysfunction tests rather against the use of antigen tests with high sensitivity and specificity than with fever testing, which have not been used widely during the latter part of the pandemic
- please also include a paragraph on the availability and feasibility of olfactory tests in contrast to other testing methods (antigen tests or pooled PCR)

Methods:

- please use antigen tests, with adequate sensitivity and specificity and rapid tests as a comparison in addition to PCR

Results:

- See above, please perform the meta-analysis mentioned above or provide adequate estimates of prevalence of olfactory symptoms not in those notified as cases but in those infected from population based studies
- accordingly please adjust the used prevalences and description of results.

Conclusions: I would be interested to see the concerns above be reflected in the conclusion. While costs are well adressed in the discussion, again I am not so sure about actual feasibiltiy of such a testing strategy. Any practice examples of this being already in place in any region or institution would be good to have.

Reviewer #2:

Remarks to the Author:

The manuscript describes how modeling the effectiveness of olfactory testing can limit SARS-CoV-2 transmission. By modeling viral transmission in a community-based setting, the authors show that olfactory testing can be a possible advantageous strategy to increase disease control. Although the idea and applied computations seems promising, the model is built on a number of assumptions, some of which may not reflect the applications in a real-world setting. Some very relevant limitations for real-life-use are raised by the authors in the last part of the discussion. However, from an olfactory perspective, I have some concerns regarding the numbers included in the model that should be addressed before publication of the model and results.

Major concerns

1) Estimate of COVID-19-independent olfactory dysfunction and author's definition of olfactory dysfunction.

The applied estimate of COVID-19-independent olfactory dysfunction (4%) is not high enough. This is also evident in the referenced papers:

REF. 29: Anosmia and hyposmia, the inability or decreased ability to smell, is estimated to afflict 3–20% of the population.

REF. 30: A the large community sample (n = 7267) who were tested with the Sniffin' Sticks Screening 12 test, olfactory dysfunction was much higher than 4%. In the referenced article, 5.1% of the participants were anosmic (score ≤ 6), 52.4% were dysosmic (7 ≤ score ≤ 10), and 42.5% were normosmic (score ≥ 11).

REF. 31: Based on objective olfactory assessment, population-based studies of olfactory loss indicate that the prevalence of smell dysfunction varies between 2.7 and 24.5% (depending on age range and other study differences). These data suggest that there is a relatively high prevalence of olfactory impairment.

Especially the large discrepancy between the modeled prevalence in the current manuscript and the relatively low prevalence of normosmic olfactory screening results in REF30 (42.5%) is concerning. This study used one of the most well-validated olfactory screening tools (Sniffin' Sticks 12 item identification test), which would be comparable to a quick screening test (4-5 minutes).

It seems the estimate used in the manuscript is related to anosmia and does not include hyposmia. However, when COVID-19 patients are tested with validated olfactory tests, many suffer from hyposmia and not anosmia [1 2].

The authors must clearly define the term 'olfactory dysfunction' – does it include hyposmia and parosmia or only anosmia? From the last part of the discussion, it seems that both anosmia and hyposmia are included in the definition, but does this also apply in the earlier part of the manuscript and the model? If only anosmia is included in the definition, the prevalence of COVID-19-related olfactory dysfunction is reduced. If hyposmia is included, the COVID-19-independent prevalence of olfactory dysfunction should be increased substantially in the model.

2) Estimated participant inclusion of 80%

In the general population, roughly 1-3% are anosmic and 15-20% are hyposmic (manuscript reference 29). These individuals cannot participate in the regular screening programme, why the estimate of 80% inclusion does not seem realistic.

Would the model still result in favourable use of olfactory testing if participation was reduced to 60-70%? Furthermore, as these individuals with pre-COVID-19 olfactory disorders could be asymptomatic SARS-COV-2 spreaders, this could have a negative impact on disease control. Can

this be incorporated into the model?

MINOR CONCERNS

1) The authors use a CDC website for reference when stating that olfactory dysfunction has no association with influenza even though the website correctly lists 'runny or stuffy nose' as a symptom of the flu which under normal circumstances causes temporary olfactory dysfunction. Many other viral infections - including common cold - are known to also cause olfactory dysfunction [3]. I would suggest rephrasing the sentence '...and no association with influenza ...' to acknowledge that olfactory loss can occur as a symptom of other viral infections (among many other causes).

2) Current knowledge on duration of olfactory dysfunction suggests that the deficit does not only last for 7 days, at least for most patients. In a recent large study, Olfactory recovery within 40 days of respiratory symptom onset was reported for ~50% of participants [4]. As such, the following statement does not reflect the normal recovery of olfactory loss: '...Fourth, COVID-induced olfactory dysfunction has been shown to last roughly 7 days in clinical studies...'. This seems to be advantageous for the modeling and could be included in the discussion.

Literature

1. Iannuzzi, L. et al. Gaining Back What Is Lost: Recovering the Sense of Smell in Mild to Moderate Patients After COVID-19. *Chem Senses* 45, 875–881 (2020).
2. Prajapati, D. P. et al. Association of subjective olfactory dysfunction and 12-item odor identification testing in ambulatory COVID-19 patients. *International Forum of Allergy & Rhinology* 42, 223–9 (2020).
3. Pellegrino, R., Walliczek-Dworschak, U., Winter, G., Hull, D. & Hummel, T. Investigation of chemosensitivity during and after an acute cold. *International Forum of Allergy & Rhinology* 7, 185–191 (2016).
- 4 Gerkin, R. C. et al. Recent smell loss is the best predictor of COVID-19 among individuals with recent respiratory symptoms. *Chem Senses* (2020). doi:10.1093/chemse/bjaa081

Reviewer #3:

Remarks to the Author:

In general,

all possible alternative infection tests are interesting in the COVID pandemic...

This one is particularly interesting, seemingly rather predictive and cheap.

Here are some comments about the various sections of the ms that the authors might want to respond to and, if they agree, introduce in the ms.

Introduction

The reported prevalence of 82% of OD in "otherwise entirely asymptomatic" may not coincide with other definitions of "asymptomatic" or "asymptomatic" and therefore the percentage of SARS2-COV infected that will not be detected by this method (False Negative) is uncertain. By definition, "totally asymptomatic infected" do not suffer from anosmia either. Furthermore, a certain percentage of (healthy) population has OD and this percentage seems to vary a lot with age, sex, nationality and other factors. Thus, the true False Negative and False Positive rates of this test (also dependent on test type...) remain unclear, although the problem is discussed in the last section before Discussion and at the end of Discussion, but the conclusion, that the test is best used on young and very young individuals, may conflict with less adherence to a self-administered and -evaluated test in these age groups.

Methods summary

In the first model, it appears that 1/3 is naturally symptomatic (and will self-isolate) and 2/3 will depend on being screened. These percentages are still uncertain but will influence the "gain in R_0 " of screening. Does the stated value of R_0 (1.5) include this naturally occurring self-isolation? Are the basic assumptions common to the second set of analyses? Is the 80% participation randomized independently each time or are the participating/non-participating individuals fixed throughout the simulations?

Results

This section shows that the proposed test has a positive effect on detection of infected, but dependent on several parameters. Is there no possibility to find an expression, even approximate, for the effects in order to reduce dependence on the chosen scenarios?

Discussion

The use of a repeated self-administered test, apart from a learning effect and the problems with self-administration, may also lead to less adherence to distancing and other preventive practices, especially among individuals with perceived negative test, thus increasing infection risk during times without the sought symptom. Could this be a problem? These problems are mentioned in the Discussion, but could they not be included in the simulations in some way?

Dear Reviewers,

We thank you for the thorough comments and suggestions for our manuscript, and for the positive overall assessment.

Our revised manuscript now includes a more detailed review of the literature, including multiple new relevant publications since our first submission, as suggested by reviewers. We also integrate the suggestion of reviewers to consider the fact that not all individuals who would want to participate in a screening program would be able to, due to impaired olfaction related to, for example, older age. As a result, our modeling now limits the effective participation rate in a screening program more than in our first submission. Importantly, we nevertheless find that screening for olfactory dysfunction would provide a powerful and complementary approach to reducing community transmission.

Below, we provide detailed responses to reviewer recommendations, and we have submitted a version of the revised manuscript with edited sections highlighted in blue text. Thank you for your continued consideration.

Sincerely,
Dan Larremore, Derek Toomre, and Roy Parker.

Reviewer #1 (Remarks to the Author):

Dear authors and editors, thank you very much for the chance to review this report.

This is an interesting modelling study assessing the effect of frequent olfactory testing on infections and Rt. It is well and concisely written and captures some of the important parts of olfactory testing in COVID.

I have a general comment: I am not certain about the high prevalence of olfactory symptoms/positive olfactory dysfunction tests assumed in those with SARS-CoV-2 infections, but this is a crucial assumption underlying the conclusions of the paper.

As far as I understand the meta-analyses presented as references for the current estimates provide prevalence of anosmia mostly in those patients notified in cases and often from highly symptomatic patients such as hospital patients and not in those infected (detected via population-based studies). In my opinion however the actual prevalence of those infected should be used as a modelling basis – which should be meta-analysed from population based studies (seroprevalence or other). I have not found such a meta-analysis but from our own data would assume that prevalence then would be lower. Also, in German contact tracing data (so notified cases) anosmia is present in 21%¹, which of course may be an underestimate. If that was the case, sensitivity of olfactory tests is far below sensitivity of current antigen or PCR tests. Currently the lower bound of sensitivity analyses provided by the authors is at 25% prevalence and does not show an effect (which would change conclusions of the paper if this was the more probably prevalence of anosmia in all infected) so I would ask authors to please

- perform/reference a meta-analysis/studies of anosmia prevalence in population based studies and parametrize the model/ change sensitivities accordingly

- change parameters and conclusions accordingly

Our revised manuscript now has an expanded and extensive discussion of the literature in the Introduction, showing that a large fraction of both symptomatic and asymptomatic COVID-19 infected individuals show olfactory dysfunction when evaluated using a standardized test. We further summarize this literature in a detailed Supplemental Table S1 that lists relevant papers, findings, sample sizes, and confidence intervals. This new table also includes studies of asymptomatic subjects and children, with rates of COVID-19 olfactory dysfunction >80% in both cases. Most of the studies, however, focused on mild to moderate severity outpatients, not highly symptomatic patients. Indeed, we now note that in severe patients the prevalence of olfactory dysfunction is lower, an observation complicated by the fact that they may be later in the clinical course of the disease at the time of observation. This expanded analysis of the

1

https://www.rki.de/DE/Content/InfAZ/N/Neuartiges_Coronavirus/Situationsberichte/Jan_2021/2021-01-19-de.pdf?__blob=publicationFile

literature allows us to be confident that a minimum of 50%, and more realistically 80%, of SARS-2-CoV infected individuals will show olfactory defects.

One further point which we also address in our revisions is to differentiate more clearly between the *prevalence* of COVID-19 related olfactory dysfunction and the *sensitivity* of a test instrument to identify it. By prevalence, we specifically mean whether, during the clinical course, the person experiences olfactory dysfunction (OD) at any point. This is a parameter which is used in the modeling. By sensitivity, this directly means how the outcomes of an olfactory test compare to the diagnosis of an RT-PCR test for SARS CoV-2, at any point in time. Because the mean duration of PCR positivity (~3 weeks) is longer than that for OD (~1-2 weeks), many individuals will be PCR positive without OD in the earliest days of infection as well as the final week of infection. Given this, the prevalence parameters we use in the manuscript are in an appropriate range, and may be slightly conservative: if the sensitivity of an OD test is 75%, the true prevalence may be higher.

Changes:

- Expanded literature review, following reviewer recommendations and new literature developments.
- New table S1 detailing key estimates of COVID-19-related OD prevalence, with uncertainty and sample size information.
- Improved wording and clarification throughout manuscript.

As a second general comment I am not convinced by the used comparison to weekly PCR and would like to see additional comparison to higher frequency use of antigen tests/pooled PCR. This is important also as the authors do point out the low positive predictive value of olfactory testing because of prevalent non-COVID related olfactory symptoms in the population.

Introduction:

- please contrast the use of olfactory dysfunction tests rather against the use of antigen tests with high sensitivity and specificity than with fever testing, which have not been used widely during the latter part of the pandemic

This is an excellent suggestion given the growing availability of relatively inexpensive antigen tests. To address this comment, we have added a comparison of olfactory testing to frequent screening using antigen tests. As we note, PCR with 1 day turnaround is comparable to rapid antigen testing because, although less sensitive than PCR, does not have the 1 day or greater delay in reporting (Larremore et al., *Science Advances*, 2020).

We hope to retain the references to fever screening in the introduction to develop the reader's intuition for the appeal of a rapid and ultra-low-cost screening symptom. Fever screening, while an utter failure in the CDC's own evaluation, continues to be widely used at hospital entrances,

in physicians' and dentists' offices, and other locations². The only reason for this continued screening is its near-zero marginal cost.

Changes:

- All OD testing regimens are now compared to both PCR (with one-day turnaround) and rapid antigen testing, with the specific characteristics of the Abbott BinaxNOW test, a relatively inexpensive and relatively high performance antigen test which has been independently evaluated³.

- please also include a paragraph on the availability and feasibility of olfactory tests in contrast to other testing methods (antigen tests or pooled PCR)

Changes:

- We have expanded our Discussion of this topic.

Methods:

-please use antigen tests, with adequate sensitivity and specificity and rapid tests as a comparison in addition to PCR

Changes:

- As noted above, we now compare to an antigen test with the performance characteristics of the BinaxNOW established during independent field tests.

Results:

- See above, please perform the meta-analysis mentioned above or provide adequate estimates of prevalence of olfactory symptoms not in those notified as cases but in those infected from population based studies
- accordingly please adjust the used prevalences and description of results.

As detailed above, we have added an extensive discussion and documentation of the literature with a new Supplemental table demonstrating high prevalence of olfactory dysfunction when individuals with COVID are tested with a standardized olfaction test.

Conclusions:

I would be interested to see the concerns above be reflected in the conclusion. While costs are well addressed in the discussion, again I am not so sure about actual feasibility of such a testing strategy. Any practice examples of this being already in place in any region or institution would be good to have.

We have tried to carefully highlight the limitations of our analysis, and the feasibility of such an approach. To our knowledge, no one has used olfactory screening on a large scale to perform

² In the experience of the authors. Candidly, the continued screening for fever at this stage of the pandemic is utterly baffling.

³ <https://academic.oup.com/cid/advance-article/doi/10.1093/cid/ciaa1890/6052342>

COVID-19 surveillance. Indeed, one reason may be that it had not yet been shown to be theoretically impactful.

Encouragingly, based on the preprint version of the present manuscript, identical to our submission to *Nature Communications*, a small private university in Switzerland has launched such a screening program for olfactory dysfunction. This parallels less formal efforts by Penn State University in the U.S. with a “daily smell check” campaign—though in the PSU program, individuals are simply encouraged to smell their morning coffee, shampoo, or other household objects to quickly check for complete anosmia, differing markedly from the quantitative and standardized tests we advocate for in this manuscript. Nevertheless, these efforts show the value of the present work in stimulating the use of olfactory dysfunction screening for COVID-19 control.

Reviewer #2 (Remarks to the Author):

The manuscript describes how modeling the effectiveness of olfactory testing can limit SARS-CoV-2 transmission. By modeling viral transmission in a community-based setting, the authors show that olfactory testing can be a possible advantageous strategy to increase disease control. Although the idea and applied computations seems promising, the model is built on a number of assumptions, some of which may not reflect the applications in a real-world setting. Some very relevant limitations for real-life-use are raised by the authors in the last part of the discussion. However, from an olfactory perspective, I have some concerns regarding the numbers included in the model that should be addressed before publication of the model and results.

Major concerns

1) Estimate of COVID-19-independent olfactory dysfunction and author's definition of olfactory dysfunction. The applied estimate of COVID-19-independent olfactory dysfunction (4%) is not high enough. This is also evident in the referenced papers:

REF. 29: Anosmia and hyposmia, the inability or decreased ability to smell, is estimated to afflict 3–20% of the population.

REF. 30: A large community sample ($n = 7267$) who were tested with the Sniffin' Sticks Screening 12 test, olfactory dysfunction was much higher than 4%. In the referenced article, 5.1% of the participants were anosmic (score ≤ 6), 52.4% were dysosmic ($7 \leq \text{score} \leq 10$), and 42.5% were normosmic (score ≥ 11).

REF. 31: Based on objective olfactory assessment, population-based studies of olfactory loss indicate that the prevalence of smell dysfunction varies between 2.7 and 24.5% (depending on age range and other study differences). These data suggest that there is a relatively high prevalence of olfactory impairment.

Especially the large discrepancy between the modeled prevalence in the current manuscript and the relatively low prevalence of normosmic olfactory screening results in REF30 (42.5%) is concerning. This study used one of the most well-validated olfactory screening tools (Sniffin' Sticks 12 item identification test), which would be comparable to a quick screening test (4-5 minutes).

It seems the estimate used in the manuscript is related to anosmia and does not include hyposmia. However, when COVID-19 patients are tested with validated olfactory tests, many suffer from hyposmia and not anosmia [1 2].

The authors must clearly define the term 'olfactory dysfunction' – does it include hyposmia and parosmia or only anosmia? From the last part of the discussion, it seems that both anosmia and hyposmia are included in the definition, but does this also apply in the earlier part of the manuscript and the model? If only anosmia is included in the definition, the prevalence of

COVID-19-related olfactory dysfunction is reduced. If hyposmia is included, the COVID-19-independent prevalence of olfactory dysfunction should be increased substantially in the model.

This feedback has led us to make numerous changes in the manuscript. First, we now clarify the issue of hyposmia and anosmia in the text by explicitly stating that, for the purposes of this modeling study, we consider olfactory dysfunction to include defects in olfaction that could be detected by a simple standardized olfaction test, including hyposmia and anosmia. However, we also note that there is no uniform definition of the exact cutoff for hyposmia, or its subcategories of mild, moderate, or severe hyposmia / micro-anosmia (e.g. Moein et al. 2020), and different device manufacturers use different cutoffs.

In addition to slightly varying definitions, there is also the issue of when, during the course of an infection, an individual is tested. For the purposes of public health screening tests, identification of COVID-19-related OD is clearly more valuable when it is earlier. However, the timing of reporting in the literature varies. For instance, lanuzzi et al [1] shows only 10% of patients were anosmic while 53% were hyposmic using an SST score, but a study from Lechien et al⁴ who also use an SST score find 48% anosmia and 14% hyposmia. The key difference: the lanuzzi study focused on hospitalized patients who were tested around 20 days after hospitalization. Similarly, Prajapati et al [2] report average scores consistent with hyposmia (7.93 out of 12 BSIT), but also report that the average time between onset of smell loss and BSIT testing was 10.9 days.

The most parsimonious explanation of the findings in [1,2] is that by days 10-20 after symptom onset, most anosmia had fully to partially resolved to hyposmia. We think that this is an important clarification, and have added language to introduce these studies and clarify this point in the text.

One additional data point, which we do not cite in the paper, comes from a recent study in Brazil using a 10 odorant test where outpatients with suspected COVID-19 (n=160) were also tested via PCR. 48% of PCR+ subjects had a score of 5/10 or lower, qualifying as severe hyposmia or anosmia. No patients had a fever. The specificity of the olfaction test was 95% in predicting PCR+ COVID-19. We suggest *not* to cite this study as it has not yet been peer reviewed, and the olfaction test was provided by author Derek Toomre (COI disclosed), but note it here as it may be relevant to the reviewer⁵—there are relatively few studies that examine specificity as this requires both PCR and olfaction testing of all individuals studied.

The reviewer raises a critical point that the degree of COVID-19 independent olfactory dysfunction will influence the effectiveness of any screening program. To address this issue, **we have re-done all analyses such that we have increased the rate of COVID-19 independent olfactory dysfunction to 20%**. Combined with our assumed 20% opt-out rate in testing, this means that only 64% of the population was assumed to either wish to or be able to participate in

⁴ <https://onlinelibrary.wiley.com/doi/pdfdirect/10.1002/hed.26279>

⁵ <https://www.medrxiv.org/content/10.1101/2021.01.20.21250173v1.full>

a screening program. An important observation is that olfactory screening is still an effective tool even under this high rate of COVID-independent olfactory dysfunction.

Changes:

- Expanded review of the literature in introduction and in a new Supplementary table.
- Clarification of precisely what is meant by olfactory dysfunction in the current study.
- Improved precision with vocabulary throughout.
- Decrease of participation rate in olfaction screening from 80% (version 1) to 64% (present version) to reflect 20% of individuals with existing COVID-19-independent olfactory dysfunction, which would exclude their participation.

2) Estimated participant inclusion of 80%

In the general population, roughly 1-3% are anosmic and 15-20% are hyposmic (manuscript reference 29). These individuals cannot participate in the regular screening programme, why the estimate of 80% inclusion does not seem realistic.

Would the model still result in favourable use of olfactory testing if participation was reduced to 60-70%? Furthermore, as these individuals with pre-COVID-19 olfactory disorders could be asymptomatic SARS-COV-2 spreaders, this could have a negative impact on disease control. Can this be incorporated into the model?

To address this suggestion, we have reduced the total participation rate to 64% (see notes above). Importantly, we find that (1) OD screening is nevertheless an effective way to reduce rates of transmission, even at 64% net participation; and (2) the effective cost of the two-step point-of-entry screening program (OD screening followed by a reflex rapid antigen test) increases, due to the increase in the number of antigen tests needed. Nevertheless, the two-step screening at points of entry is predicted to be highly effective and specific.

MINOR CONCERNS

1) The authors use a CDC website for reference when stating that olfactory dysfunction has no association with influenza even though the website correctly lists 'runny or stuffy nose' as a symptom of the flu which under normal circumstances causes temporary olfactory dysfunction. Many other viral infections - including common cold - are known to also cause olfactory dysfunction [3]. I would suggest rephrasing the sentence '...and no association with influenza ...' to acknowledge that olfactory loss can occur as a symptom of other viral infections (among many other causes).

This is a good point and we have rephrased the text to note that both other infections, as well as chronic conditions such as Parkinson's or Alzheimer's Disease can cause olfactory dysfunction. Indeed, the fact that olfactory dysfunction can be caused by other perturbations or infections is the reason we require follow-up (reflex) molecular testing in the olfactory surveillance screening approach.

Changes:

- Expanded discussion of other causes of olfactory dysfunction.

2) Current knowledge on the duration of olfactory dysfunction suggests that the deficit does not only last for 7 days, at least for most patients. In a recent large study, Olfactory recovery within 40 days of respiratory symptom onset was reported for ~50% of participants [4]. As such, the following statement does not reflect the normal recovery of olfactory loss: ‘...Fourth, COVID-induced olfactory dysfunction has been shown to last roughly 7 days in clinical studies...’. This seems to be advantageous for the modeling and could be included in the discussion.

The reviewer is correct that olfactory dysfunction can last longer than 7 days. We have adjusted the text to note this issue. Papers [1,2] in which testing revealed persistent hyposmia after 10 days and 20 days supports this conclusion. However, because individuals with COVID are only infectious for around 1 week, we have not expanded our modeling beyond 7 days as this would not be productive in identifying infectious individuals. More succinctly: in a screening program, any prolonged olfactory dysfunction would be irrelevant since the individual would have been detected early when olfaction was first compromised.

Changes:

- Noted the potential long duration of olfactory dysfunction.

Literature

1. Iannuzzi, L. et al. Gaining Back What Is Lost: Recovering the Sense of Smell in Mild to Moderate Patients After COVID-19. *Chem Senses* 45, 875–881 (2020).
2. Prajapati, D. P. et al. Association of subjective olfactory dysfunction and 12-item odor identification testing in ambulatory COVID-19 patients. *International Forum of Allergy & Rhinology* 42, 223–9 (2020).
3. Pellegrino, R., Walliczek-Dworschak, U., Winter, G., Hull, D. & Hummel, T. Investigation of chemosensitivity during and after an acute cold. *International Forum of Allergy & Rhinology* 7, 185–191 (2016).
- 4 Gerkin, R. C. et al. Recent smell loss is the best predictor of COVID-19 among individuals with recent respiratory symptoms. *Chem Senses* (2020). doi:10.1093/chemse/bjaa081

Reviewer #3 (Remarks to the Author):

In general, all possible alternative infection tests are interesting in the COVID pandemic... This one is particularly interesting, seemingly rather predictive and cheap.

Here are some comments about the various sections of the ms that the authors might want to respond to and, if they agree, introduce in the ms.

Introduction

The reported prevalence of 82% of OD in "otherwise entirely asymptomatic" may not coincide with other definitions of "asymptomatic" or "asymptomatic" and therefore the percentage of SARS2-COV infected that will not be detected by this method (False Negative) is uncertain. By definition, "totally asymptomatic infected" do not suffer from anosmia either.

Thank you for these supportive remarks. We might note that a test that is fast, inexpensive, and could be used frequently may have a high impact especially in lower income countries that do not have the infrastructure for complex molecular tests, funding resources are tight, and widescale vaccination is still 6-18 months away.

We completely agree that the use of "asymptomatic" to describe someone with olfactory dysfunction is confusing. The study of Bhattacharjee et al (Lancet EClinical Medicine, 2020) was done at a time in India when the government did not yet recognize "new loss of smell or taste" as a symptom of COVID-19, so, rather confusingly, some of those who were categorized as asymptomatic but nevertheless had olfactory dysfunction as a solitary symptom. 5 of 33 self-reported loss of smell as the solitary symptom, and a further 22 suffered olfactory dysfunction but did not self-report any symptoms (and we agree that these 5 should be excluded as asymptomatic).

To cover the range of possible values for the prevalence of olfactory dysfunction as a test-recognized symptom, we continue to consider symptom prevalence estimates ranging from a low 25% to a high 90%, with 50% (conservative) and 75% (best estimate) values as well. However, we have also made the following changes.

Changes:

- We have adjusted the test to more clearly differentiate between truly asymptomatic (no reported symptoms) and those who have no symptoms *except* for their self-perception of olfactory dysfunction. Critically, both groups have been shown to be identifiable using standardized OD screening tests.
- We modify the reported prevalence of "82%" of OD in otherwise entirely asymptomatic to cite the value at "79% (22 of 28)" to exclude the 5 people that self-recognize their loss of smell (and might therefore act on it).

Furthermore, a certain percentage of (healthy) population has OD and this percentage seems to vary a lot with age, sex, nationality and other factors. Thus, the true False Negative and False Positive rates of this test (also dependent on test type...) remain unclear, although the problem is discussed in the last section before Discussion and at the end of Discussion, but the conclusion, that the test is best used on young and very young individuals, may conflict with less adherence to a self-administered and -evaluated test in these age groups.

This is a good point, which was also raised by other reviewers. We now more openly discuss the fact that there may be age-correlated differences in the effectiveness of screening regimens, but more importantly, we now have decreased the effective participation rate in the screening programs: previously we considered that only 80% of people would opt to participate in screening, but we now further consider that 20% of individuals may suffer from COVID-19-independent olfactory dysfunction. As a result, we cap the fraction of the effectively participating population at 64%.

Changes:

- Net participation rate in OD screening reduced to 64% (from 80%).
- Note: participation rate in non-OD testing (PCR or antigen testing) modeled at 80%.

Methods summary

In the first model, it appears that 1/3 is naturally symptomatic (and will self-isolate) and 2/3 will depend on being screened. These percentages are still uncertain but will influence the "gain in R_0 " of screening. Does the stated value of R_0 (1.5) include this naturally occurring self-isolation? Are the basic assumptions common to the second set of analyses? Is the 80% participation randomized independently each time or are the participating/non-participating individuals fixed throughout the simulations?

Yes, the stated value of R_0 does include the naturally occurring self isolation, so that the R_0 realized by simulations reflects behavior and transmission in the absence of any additional interventions. This is detailed in the section on the calibration of R_0 , and is common to all simulation analyses in the paper.

For participation, the 64% (formerly 80% for OD screening, and still 80% for PCR or anosmia screening) are fixed ahead of time, such that a person who refuses to participate at the start of a simulation persists in their refusal throughout. Though not shown in the present manuscript, this assumption has been investigated by author Larremore, with the finding that its relaxation (so that refusal/participation happens on a test-by-test basis) does not make a meaningful impact on study conclusions.

Changes:

- Additional clarification in methods.

Results

This section shows that the proposed test has a positive effect on detection of infected, but dependent on several parameters. Is there no possibility to find an expression, even approximate, for the effects in order to reduce dependence on the chosen scenarios?

Given the complexities, we have been unable to derive a simple equation that would allow the impact of olfaction screening on R_0 to be easily predicted. One reason for this is that the parameters of frequency of testing, and the prevalence, duration, and onset of symptoms interact non-linearly. A basic regression approach to derive a simple statistical model of the impacts was fruitless, as it failed to capture those nonlinearities. We prefer to not further attempt that derivation in this manuscript.

Discussion

The use of a repeated self-administered test, apart from a learning effect and the problems with self-administration, may also lead to less adherence to distancing and other preventive practices, especially among individuals with perceived negative test, thus increasing infection risk during times without the sought symptom. Could this be a problem? These problems are mentioned in the Discussion, but could they not be included in the simulations in some way?

The learning effect is a good point, and one olfactory test manufacturer uses a training card and App to facilitate self-administration.

We agree the use of any frequent surveillance/screening test may lead to less adherence to other means of infection control. This is equally true of PCR and antigen tests, and for that matter even temperature-based screens. In revising the manuscript, we have given this issue more emphasis in the discussion.

Changes:

- Additional discussion.

Reviewers' Comments:

Reviewer #2:

Remarks to the Author:

Thank you for addressing the previous comments in parts of the manuscript. However, a very low prevalence of olfactory dysfunction (3-5% in the general population) is still indicated in the limitations section. Please add more updated information on the occurrence of olfactory dysfunction, as the incidence is higher also in the younger population:

Landis BN, Konnerth CG, Hummel T. A study on the frequency of olfactory dysfunction. *Laryngoscope* 2004;114:1764–1769.

This is also evident in larger normative studies, where younger age groups often have a higher proportion of hyposmia:

Schriever, V. A., Croy, I., Hähner, A. & Hummel, T. Updated Sniffin' Sticks normative data based on an extended sample of 9139 subjects. *Eur Arch Otorhinolaryngol* 276, 719–728 (2019).

Reviewer #3:

Remarks to the Author:

I have now read the reviewed version of the ms and find that the authors have answered all the raised questions in a satisfactory manner. I have no further comments.

Reviewer #2 (Remarks to the Author):

Thank you for addressing the previous comments in parts of the manuscript. However, a very low prevalence of olfactory dysfunction(3-5% in the general population) is still indicated in the limitations section. Please add more updated information on the occurrence of olfactory dysfunction, as the incidence is higher also in the younger population:

Landis BN, Konnerth CG, Hummel T. A study on the frequency of olfactory dysfunction. *Laryngoscope* 2004;114:1764–1769.

This is also evident In larger normative studies, where younger age groups often have a higher proportion of hyposmia:

Schriever, V. A., Croy, I., Hähner, A. & Hummel, T. Updated Sniffin' Sticks normative data based on an extended sample of 9139 subjects. *Eur Arch Otorhinolaryngol* 276, 719–728 (2019).

We have updated our manuscript to refer to both the Landis *et al.* and Schriever *et al.* papers suggested, and have further adjusted the language used to describe the results of studies in children.

Reviewer #3 (Remarks to the Author):

I have now read the reviewed version of the ms and find that the authors have answered all the raised questions in a satisfactory manner. I have no further comments.

We thank the reviewer for multiple rounds of comments, and for this positive assessment!